The effect of habitual and experimental antiperspirant and deodorant product use on the armpit microbiome

Urban Julie 1
Fergus Daniel J. 1
Savage Amy M. 2
Ehlers Megan 1 3
Menninger Holly L. 3
Dunn Robert R. 4 7
Horvath Julie E. 1 5 6 julie.horvath@naturalsciences.org
1 North Carolina Museum of Natural Sciences , Raleigh, NC , USA
2 Department of Biology & Center for Computational & Integrative Biology, Rutgers, The State University of New Jersey—Camden , Camden, NJ , USA
3 Department of Biological Sciences, North Carolina State University , Raleigh, NC , USA
4 Department of Applied Ecology and Keck Center for Behavioral Biology, North Carolina State University , Raleigh, NC , USA
5 Department of Biological and Biomedical Sciences, North Carolina Central University , Durham, NC , USA
6 Department of Evolutionary Anthropology, Duke University , Durham, NC , USA
7 Center for Macroecology, Evolution and Climate, Natural History Museum of Denmark, University of Copenhagen , Copenhagen , Denmark
Josenhans Christine
Electronic publication date: 2016 Feb 2
Publication date: 2016
Volume: 4
Electronic Location ID: e1605
Received 2014 Dec 8; Accepted 2015 Dec 27
Copyright: ©2016 Urban et al.
Copyright year: 2016
Copyright holder: Urban et al.
License: This is an open access article distributed under the terms of the Creative Commons Attribution License, which permits unrestricted use, distribution, reproduction and adaptation in any medium and for any purpose provided that it is properly attributed. For attribution, the original author(s), title, publication source (PeerJ) and either DOI or URL of the article must be cited.
License URL: https://creativecommons.org/licenses/by/4.0/

Keywords: Skin microbiome, Armpit, Axillary region, Antiperspirant, Deodorant, Skin bacteria, Microbiology

Funding: NSF #0953390 #1319293 Army Research Office W911NF-14-1-0556 Howard Hughes Medical Institute #52006933 This project was supported by an NSF Career grant #0953390 and Army Research Office grant W911NF-14-1-0556 awarded to RRD, and a Howard Hughes Medical Institute grant #52006933 through the Precollege and Undergraduate Science Education Program (http://www.hhmi.org/grants/office/undergrad/) awarded to NC State University. DJF and HLM were supported by NSF grant #1319293 awarded to RRD. The funders had no role in study design, data collection and analysis, decision to publish, or preparation of the manuscript.

==============================
An ever expanding body of research investigates the human microbiome in general and the skin microbiome in particular. Microbiomes vary greatly from individual to individual. Understanding the factors that account for this variation, however, has proven challenging, with many studies able to account statistically for just a small proportion of the inter-individual variation in the abundance, species richness or composition of bacteria. The human armpit has long been noted to host a high biomass bacterial community, and recent studies have highlighted substantial inter-individual variation in armpit bacteria, even relative to variation among individuals for other body habitats. One obvious potential explanation for this variation has to do with the use of personal hygiene products, particularly deodorants and antiperspirants. Here we experimentally manipulate product use to examine the abundance, species richness, and composition of bacterial communities that recolonize the armpits of people with different product use habits. In doing so, we find that when deodorant and antiperspirant use were stopped, culturable bacterial density increased and approached that found on individuals who regularly do not use any product. In addition, when antiperspirants were subsequently applied, bacterial density dramatically declined. These culture-based results are in line with sequence-based comparisons of the effects of long-term product use on bacterial species richness and composition. Sequence-based analyses suggested that individuals who habitually use antiperspirant tended to have a greater richness of bacterial OTUs in their armpits than those who use deodorant. In addition, individuals who used antiperspirants or deodorants long-term, but who stopped using product for two or more days as part of this study, had armpit communities dominated by Staphylococcaceae, whereas those of individuals in our study who habitually used no products were dominated by Corynebacterium. Collectively these results suggest a strong effect of product use on the bacterial composition of armpits. Although stopping the use of deodorant and antiperspirant similarly favors presence of Staphylococcaceae over Corynebacterium, their differential modes of action exert strikingly different effects on the richness of other bacteria living in armpit communities.

Introduction

Like the gut or the mouth, the human skin is covered with life. This life includes bacteria, fungi, Archaeans, bacteriophages, and even animals such as nematodes and Demodex mites (Marples, 1965; Grice & Segre, 2011; Kong & Segre, 2012). Since the 1950s it has been clear that the precise composition of the skin biome influences its effectiveness as a defensive layer against pathogens (Eichenwald et al., 1965), and contributes to bodily odors (Shelley, Hurley & Nicholas, 1953). Some species are better at defending our skin than others (Christensen & Brüggemann, 2014), just as some species produce different odors than do others (Leyden et al., 1981). What is unclear is the extent to which human behaviors influence the composition of skin microbes. Inasmuch as two types of products, antiperspirants and deodorants, are used daily in armpits by a large number of people (perhaps as many as 90% of people in the US, according to Benohanian, 2001), armpits represent an interesting context in which to explore the general phenomenon of how human behavior and product use influence skin microbes.

A long history of work focuses on the biology of the culturable bacteria in armpits (Shelley, Hurley & Nicholas, 1953; Marples, 1965; Leyden et al., 1981). More recent work has built upon this history to consider both those taxa that are culturable and those whose presence is only detectable (to date) through sequencing. In one of the first of this new wave of studies, Grice et al. (2009) sampled, cloned, and Sanger-sequenced bacteria from 20 body regions sampled from nine participants. The most prominent bacteria present in armpits were species of Corynebacterium, Staphylococcus, Betaproteobacteria, Clostridiales, Lactobacillus, Propionibacterium, and Streptococcus. Interestingly, bacterial residents of armpits were shown to be highly variable even across this small number of participants: four participants’ communities were dominated by Corynebacterium species, three by Staphylococcus species, and two by Betaproteobacteria. Gao et al. (2010) also found large variation among individuals in the composition of armpit bacterial taxa (drawn from similar genera as in Grice et al., 2009). This high person-to-person variability stands in contrast to samples from other skin habitats, which show less inter-individual variation, and are locations where product use is less common (Costello et al., 2009; Caporaso et al., 2011; Hulcr et al., 2012) (although see Callewaert et al., 2013). This variability might simply reflect stochastic effects or even the sequencing of dead bacteria on the skin (Grice & Segre, 2011). However, Egert et al. (2011) found that most of the same common taxa in the armpits, including most/all of the common taxa found in Grice et al. (2009) and Gao et al. (2010), were the most metabolically active and contributed the most to armpit odor.

The high variability in armpit communities among individuals suggests that an unaccounted for factor, perhaps product use, might be exerting a strong influence on armpit bacteria, which may in turn have functional consequences for the host. Older, culture based studies suggest that the use of deodorants and antiperspirants appears to reduce the abundance of culturable bacterial taxa, particularly those of Corynebacterium, a slow-growing lineage of bacteria that plays a key role in the production of armpit odor (Leyden et al., 1981). This effect is not accidental inasmuch as the intent of underarm products has long been the reduction in armpit odor either through direct reduction in the biomass of bacteria, or through blocking the exudates of the apocrine glands, which become odiferous when metabolized by bacteria (Taylor et al., 2003; Wilke et al., 2007; Fredrich et al., 2013). Second, two recent studies by Callewaert et al. (2013) and Callewaert et al. (2014) found an association between product use and the diversity of bacteria in armpits. Although the Callewaert et al. (2014) study that specifically tested for effects of product use in nine people did not include a control group (people who habitually do not use any products), this work is clearly suggestive of a potentially large impact of underarm products on entire communities of armpit bacteria.

Here, we examine several questions related to product use and armpit bacterial communities. First, we test whether there is a direct relationship between the abundance of readily culturable bacteria and product use. Second, we use a sequencing based approach to consider whether there are long-term differences in the species richness and composition of armpit bacteria on individuals who habitually use antiperspirant, deodorant or no product. We also consider whether these differences are in line with what would be expected given the intended effects of underarm products in reducing the abundance of odor-causing armpit bacteria (i.e., primarily Corynebacterium), as well as the different mechanisms by which antiperspirants vs. deodorants achieve this reduction. Finally, we compare the abundance of two focal taxa, an OTU categorized as Staphylococcaceae (predominant in comparison to the OTU categorized as Staphylococcus) and an OTU of Corynebacterium, as a function of product use, gender (based on Callewaert et al.’s, 2013 work), and time since ceasing product use.

Materials and Methods

Participants

Eighteen individual citizen scientists (public participants in scientific research) were recruited through the NC Museum of Natural Sciences’ Genomics & Microbiology Lab for armpit community sampling. Prior to the start of the experiment, the proposed study was reviewed and approved by the North Carolina State University Institutional Review Board for the Use of Human Subjects in Research (IRB#1987). All participants were supplied with an IRB authorized consent form and all provided their consent to participate (indicated by their signatures) prior to the start of the experiment. Individuals were recruited to represent three groups, each with equal numbers of men and women. One group of participants reported typically not using any deodorants or antiperspirants. Another group reported regular use of deodorant-only products. The third group reported regular use of antiperspirant products. Not everyone in our study used the same product brand, but all antiperspirant users used products containing aluminum zirconium trichlorohydrex Gly as the active ingredient. Although we designed our study to have equal group sizes, participant drop-out and product mis-classification during the course of the study resulted in a final sample of five participants each in the no product use (three men and two women) and deodorant-only use (three men and two women) groups, and seven participants (three men and four women) who used antiperspirant-containing products.

We experimentally altered product use by the participants over eight days. On the first day (Day 1), participants went about their normal habits (e.g., applying deodorant or antiperspirant if they normally wore it); we did not require that they apply product at any certain time of day or number of times per day, nor did we require that they shower a certain amount. All participants indicated showering/bathing 3–14 times per week (Table S1), and they were asked to continue their normal showering/bathing routine for this experimental week. On Days 2–6, participants were asked to discontinue product usage. During the last two days (Days 7 and 8) of sampling, all individuals, including those who did not normally use antiperspirant or deodorant, were asked to use an antiperspirant/deodorant product (we provided Secret Powder Fresh for women and Old Spice Fiji for men—both contained aluminum zirconium trichlorohydrex Gly as the active ingredient). On each of the eight days of the study, both armpits of each individual were swabbed once for 45–60 s between 11 am (EST) and 1 pm (EST) with a dual-tipped sterile BBL™ CultureSwab™ (Becton Dickinson and Company, Franklin Lakes, NJ, USA). A tradeoff exists between sampling participants many times (daily) over a short period of time and sampling less frequently over a longer period of time (weeks to months). We opted for more frequent sampling (daily for eight days) vs. longer-term sampling to assess shorter-term effects of ceasing product use in order to achieve a balance across competing considerations including participant compliance, available budget, supplies and personnel time.

Culture-based sampling

Bacteria sampled from the left and right armpits were cultured the same day they were collected by immersing one of the two sample swabs into 0.5 mL of phosphate buffered saline (PBS), mixing, and spreading 20 µl of this solution onto sterile LB agar plates. All cultures were incubated aerobically at 37 °C for approximately 22 h, and then stored at 4 °C to stop further colony growth. Plates were photographed and numbers of colonies occupying a standard central region of each plate (18.07 cm2) were counted using Image J software (version 1.46, Rasband, 1997–2014). The 18.07 cm2 region on the plate does not necessarily equate to the same size region on a person’s skin. Abundance counts for bacteria cultured from the left vs. right armpit were averaged to yield one abundance score (i.e., average number of CFU present on two culture plates) per person for each day of the experiment.

Culture-based analyses

To test the effects of time and change in product use on abundance of culturable bacteria, we ran several analyses of variance (ANOVA) on time intervals of interest, using SPSS version 21.0 (IBM Inc., Armonk, NY, USA). To determine whether regular use of underarm products exerts an effect on initial abundance immediately following disuse of products, we ran a one-way ANOVA on abundance of culturable bacteria sampled on Day 2 (the first day all subjects ceased use of products). We conducted mixed-model (day × product use × gender) ANOVAs to test effects of stopping product use and continued disuse on abundance of culturable bacteria (Day 1 vs. Day 2–6), and to test effect of antiperspirant application on abundance (Day 6 vs. Days 7–8).

Sequence-based sampling

Several extensive surveys of armpit microbes (e.g., Flores, Henley & Fierer, 2012) and our personal observations indicate that armpit samples show poor amplification for sequence-based analyses. In addition, our preliminary experiments indicated that presence of product (antiperspirant or deodorant) inhibited our ability to isolate and subsequently PCR amplify armpit bacterial DNA, so we chose to conduct our “early” sequence-based sampling on Day 3 (the second day of continued product disuse) and the “late” sampling on Day 6 (the fifth and final day of continued product disuse). The expectation was that on Day 3, residual effects of product use would remain persistent (if they existed) and that by Day 6 bacterial populations might have begun to recover from whatever such effects might be, and hence converge on a common, shared composition.

DNA was isolated from the second sample swab, stored dry at −20 °C for up to one month prior to isolation, using the PowerSoil DNA Isolation Kit (MO BIO Laboratories, Carlsbad, CA, USA) with minor modifications. Instead of using soil, the swab tip was swirled into the beads for approximately 5 s and removed before adding solution C1. For the elution, solution C6 was heated to 50 °C before being added in the final step of the protocol. Also, only 50 µl of solution C6 was used to elute the DNA. All isolated DNA samples were stored at −20 °C. The V4 region of 16S rDNA was PCR amplified from each DNA sample using Premix ExTaq (Takara Bio) with primers designed to amplify bacteria and Archaea (515F: GTGCCAGCMGCCGCGGTAA, and 806R: GGACTACHVGGGTWTCTAAT), modified to include Roche 454 adapters and index sequences as previously described (Hulcr et al., 2012). Reactions were set up such that there was a unique index for each subject, arm (left vs. right) and day. All PCRs, including a no-template control, were performed in triplicate and after thermocycling, each triplicate was pooled and purified using an UltraClean-htp 96-well PCR Clean-up kit (MO BIO Laboratories, Inc., Carlsbad, CA, USA). The purified PCR products were quantified with a Qubit 2.0 and dsDNA BR Assay Kit (Invitrogen, Grand Island, NY, USA) and an equal mass (110 ng) of each was mixed into a single pool of all individuals and days. The no-template control DNA was below detectable levels and thus the entire no-template reaction mixture was added to the pool to be sequenced, to allow for detection of possible contaminants (Salter et al., 2014). An ethanol precipitation was performed to concentrate the mixed pool of index products. The DNA was sent to Selah Genomics (Greenville, SC, USA) for Roche 454 next-generation pyrosequencing. Sequence data from both axillae at day 3 are deposited in NCBI as Bioproject PRJNA281417(http://www.ncbi.nlm.nih.gov/bioproject/).

The resulting sequences were analyzed using the QIIME (version 1.7.0) microbial community analysis software (Caporaso et al., 2010). This included initial filtering of DNA sequences using default parameters (to insure their minimum and maximum lengths were 200 bp and 1,000 bp respectively, that quality score was at least 25, and that no ambiguous or mismatched bases appeared in the primer sequence) and assignment of multiplexed reads to samples based on their indexed barcode. Sequences from each sample were clustered into Operational Taxonomic Units (OTUs) based on 97% sequence similarity. Using a strategy of de novo OTU picking in QIIME, a representative sequence was picked for each OTU and taxonomy was assigned using the uclust consensus taxonomy classifier. Consistent with previous studies (e.g., Hulcr et al., 2012) and with the default in QIIME, we assigned taxonomy to Level 6 (L6), which assigns OTUs to genus level. Before performing any further analyses, we rarefied our data to 1,000 reads per armpit sample. Four samples (only a single sample from each of four people distributed across categories of product use, gender, and day) had fewer than 1,000 reads and were excluded from further analyses. In order to retain all participants in our study without having unbalanced data, we randomly selected either left or right armpit for each person and analyzed one armpit for day 3 and the same armpit for day 6. Random selection was done using the RANDBETWEEN function in Microsoft Excel 2013, and selection of right vs. left armpit for each person is included in Table S2. All subsequent sequence based analyses were based on one armpit per person. Read counts of each OTU were exported as a matrix for subsequent analyses after the removal of singletons and doubletons from the dataset.

Sequence-based analyses

For all participants we used day 3 and day 6 (for person T we had to use day 5 sample as no day 6 sample existed) to assess richness (i.e., number of OTUs) and composition at the level of each individual (number of OTUs per individual).

Richness

The program Primer-E v.6.1.15 (PRIMER-E, Plymouth, UK) was used to calculate OTU richness (defined as number of OTUs present per person per day) based on the OTU table. A mixed model ANOVA was performed using SPSS (version 21.0) to determine the effect of day (early-day 3 vs. late-day 6, as described above), and product use (regular antiperspirant users, deodorant users, or participants who did not regularly use underarm products) on richness. The relationship between the abundance of culturable bacteria and richness was computed across all groups using a Pearson rank correlation.

Composition

We visualized the composition of armpit bacteria using non-metric multidimensional scaling ordination (NMDS) in Primer-E v.6.1.13 with PerMANOVA ext. 1.0.3 (Clarke & Gorley, 2006). To do this, we first constructed NMDS plots with 100 restarts and a Type I Kruskal fit scheme based on a Dissimilarity matrix of Bray-Curtis distances. To assess the relationship between product use and sampling period (early vs. late), we conducted a permuted multivariate analysis of variance (PerMANOVA) test with treatment group (product use categories described above) and sampling period and their interaction as factors, 9,999 iterations and Type III sums of squares. We conducted SIMPER analyses for each significant factor to determine the OTUs that contributed the most to pairwise between-group differences in ordination space (Table 1). For a full comparison of all OTUs that differed in abundance between the usage groups, we conducted a Metastats analysis (see http://metastats.cbcb.umd.edu/detection.html and White, Nagarajan & Pop, 2009) using pairwise comparisons of average sequence reads for each product use group. Because Corynebacterium and Staphylococcus have been previously shown to be important taxa in armpit communities (see above), we conducted additional analyses of their abundances (i.e., number of reads) using a mixed model ANOVA with the independent factors of product use, gender, day, and their interaction. Finally, we determined the relationship between the abundance of Corynebacterium and Staphylococcus across all groups using a Spearman rank correlation. We used SAS v.9.3 Statistical Software (SAS Institute, Inc., Cary, NC, USA) to conduct both of these analyses.

Table 1 Comparisons of the average abundances and % contributions of the top 5 microbes that contributed the most to differences between each pair of treatment groups.

			Average abundance		
Comparison	Family	Genus	Antiperspirant	Deodorant	% contribution to differences	
Antiperspirant vs. deodoranta	Staphylococcaceae	Other	600.36	609.00	32.91	
Corynebacteriaceae	Corynebacterium	141.29	293.80	27.69	
Clostrididaceae	Anaerococcus	37.07	47.60	8.41	
Alicyclobacillaceae	Alicyclobacillus	26.57	4.30	3.41	
Clostrididaceae	Finegoldia	0.86	16.00	2.05	
			None	Deodorant		
None vs. deodorant	Staphylococcaceae	Other	212.90	609.00	43.63	
Corynebacteriaceae	Corynebacterium	615.90	293.80	35.36	
Clostridiaceae	Anaerococcus	78.50	47.60	8.74	
Clostridiaceae	Finegoldia	9.90	16.00	2.16	
Campylobacteraceae	Campylobacter	8.60	0.90	0.94	
			None	Antiperspirant		
None vs. antiperspirant	Corynebacteriaceae	Corynebacterium	615.90	141.29	37.82	
Staphylococcaceae	Other	212.90	600.36	34.21	
Clostrididaceae	Anaerococcus	78.50	37.07	6.98	
Alicyclobacillaceae	Alicyclobacillus	5.10	26.57	2.02	
Porphyromonadaceae	Porphyromonas	2.70	14.07	1.22	
Notes.

a In a post-hoc test, participants ceasing antiperspirant use and participants ceasing deodorant use were not significantly different from each other (P = 0.095).

Results

Culture-based results

If antiperspirants and deodorants negatively affect the abundance of bacteria in armpits, we would expect that on the first day of no product use (Day 2 in our experiment) the residual effects of products would lead to lower bacterial abundances in armpits of those individuals who use such products. As time progressed, we expect those abundances to increase among habitual product users (as bacterial populations recover). Finally, once antiperspirant was applied to (rebounded) armpit assemblages after five days of no product use, we expect abundances to decline. In order to focus on one experimental treatment (product vs. no product use), we instructed participants to alter only that aspect of their behavior.

Day 2—A significant effect of product use habit was observed on Day 2, the first day none of the subjects applied product (F2,16 = 3.9, p < 0.046; Table 2). Subsequent tests performed using Tukey HSD (Honestly Significant Difference) indicated that antiperspirant users who ceased using product initially had significantly lower abundances of bacterial colonies on culture plates than deodorant users who ceased using product (p < 0.05). Significant pairwise differences were not observed between antiperspirant users and subjects using no product, nor between either product-use group and subjects who use no product.

Table 2 Mean (standard deviation) of abundance of culturable bacteria (CFU/cm2) on culture plates at baseline (Day 1, when all subjects followed regular habit of product use application) and subsequent days (Days 2–6) when all subjects ceased usage of underarm products and (Days 7 and 8) when all subjects used underarm product provided by us.

A significant effect of product use was shown on Day 2 (F2,16 = 3.9, p = 0.046).

	Day 1	Day 2	Day 3	Day 4	Day 5	Day 6	Day 7	Day 8	
Antiperspirant users (4F, 3M)	103.1 (179.2)	158.9 (95.8)	200 (303.4)	460.4 (757.1)	902.6 (703.4)	555 (555.1)	2.7 (4.1)	33.2 (62.1)	
Deodorant users (2F, 3M)	430.3 (793.7)	645.9 (470.2)	1053.4 (1087.1)	1161.4 (798)	1033.5 (811.1)	1149.2 (923.5)	229.3 (363.4)	39.4 (60.0)	
No product users (2F, 3M)	179 (201.2)	436.9 (297.1)	597.4 (383)	704 (619.1)	600 (476.8)	626.6 (317.2)	0.8 (1.1)	2.1 (3.9)	
All groups combined	221.7 (447.2)	383.9 (353.9)	567.9 (707.0)	738.2 (748.4)	852.1 (661.9)	750.8 (652.0)	73.0 (217.0)	25.4 (48.0)	
Notes.

Abundances on Day 5 were significantly higher than on Days 1 and 2. Abundances on Day 4 were significantly higher than on Day 1. Abundances were measured by resuspending skin microbes in PBS solution and then plating 1/25th of the volume on culture plates. Values in the table are counted CFUs multiplied by 25 to represent the total number of CFUs from each sample.

Days 1–6—With the disuse of antiperspirants and deodorants, bacterial abundance significantly increased during the course of our experiment (F5,55 = 4.92, p = 0.001) (Table 2). Subsequent tests performed using Tukey HSD indicated that abundances on Day 5 were significantly higher than on Day 1 (p < 0.05) and Day 2 (p < 0.05). Abundances on Day 4 were significantly higher than on Day 1 (p < 0.05). This increase over time was independent of product use, gender, or any interactions among these variables.

Antiperspirant application (Day 6 vs. 7–8)—Bacterial abundances on Day 7 (mean CFU/cm2 ± standard deviation = 73 ± 217) and Day 8 (mean = 25.4 ± 48), after antiperspirant application, were an order of magnitude lower than on Day 6 (mean = 750.8 ± 652; overall F2,28 = 19.0, p < 0.001).

Sequence-based results

To understand the association between long-term use of antiperspirant or deodorant on more complete armpit bacterial communities (rather than just the abundance of culturable taxa), we analyzed the richness (i.e., OTU diversity) and composition of armpit microbes detected via sequencing of 16S rDNA in participants sampled on two days: early (Day 3 of the experiment, the second day of continued product disuse) and late (Day 6, the fifth day of continued product disuse). On both of these days, the only differences we expected were those due to long-term product use, which might occur due to differences in who chooses to use deodorant or antiperspirant (with individuals with more odorous microbial assemblages perhaps more likely to use product) or due to the product use itself. If the former, we would expect individuals with more product use to tend to be the same individuals with more slow-growing odor producing bacteria such as corynebacteria. If the latter, then we expected the reverse.

Richness

Before rarefaction, the pyrosequencing output yielded 133,098 reads that passed the quality screens of the 454 platform and QIIME filtering. After rarefaction to 1,000 reads per sample we observed a total of 106 OTUs of bacteria and Archaea in armpits of the 17 individuals in our study, with an average richness of 22 OTUs per person. Because Archaea were represented by just two OTUs found on the same person (Candidatus nitrososphaera and Halococcus), we hereafter focus on bacterial results.

Samples of bacteria from regular antiperspirant users, two and five days after stopping underarm product use, were more diverse (mean number of OTUs ± SD = 31.2 ± 24.4) than those of deodorant users (mean ± SD = 10.7 ± 6.2), or users of no product (mean ± SD = 20.5 ± 13.4) (ANOVA, F2,14 = 3.91, p = 0.045, Fig. 1, Tables S1 and S3). Subsequent Tukey HSD tests supported our finding that bacterial communities of antiperspirant users were significantly richer than those of deodorant users (p < 0.05). No significant differences were observed between bacterial richness of either group of product-users and users of no product. Neither a significant effect of day nor an interaction was observed. The number of OTUs was not correlated with the abundance of culturable bacteria (r = − 0.37, p > 0.05).

Figure 1 Mean composition and richness of bacterial OTUs for all three product user types, combined OTU data from two and five days after stopping product use.

Bacteria with greater than 10 sequence reads across all users in each category are shown. The top three bacterial OTUs are shown; a full list is available in Table S1. Antiperspirant users have much richer armpits (22% other bacteria versus 5% for deodorant users and 9% for no product users). At the L6 level of OTU assignment, the OTU for the highly abundant Staphylococcaceae was “Staphylococcaceae˙other” indicating that the genus was unassigned. We refer to this OTU for simplicity throughout the remainder of the figures and text as Staphylococcaceae, but that this does represent one group within Staphylococcaceae and does not denote all identified OTUs in this family.

Composition

The composition of bacteria was strongly associated with underarm product use (PerMANOVA, PTreatment = 0.0001, Fig. 2), but not sampling period (where compositional differences were measured as a function of the relative abundance of taxa based on read number). The five bacterial OTUs that contributed most to differences between each pair of product use groups based on a SIMPER analysis are shown in Table 1. These results were largely consistent with those from a Metastats analysis (White, Nagarajan & Pop, 2009), which indicated some OTUs were significantly more abundant in specific product use groups (Table S4). Overall, the bacteria that contributed the most to differences between microbial assemblages among product use groups were an OTU of Staphylococcaceae (although at the L6 level this OTU was classified as “Staphylococcaceae˙other,” we expect it almost certainly is a Staphylococcus) and an OTU of Corynebacterium. The common OTU of Staphylococcaceae was reduced in participants who did not use underarm products compared to either deodorant users (who had >186% more of the Staphylococcaceae OTU; Table 1) or antiperspirant users (who had >181% more of the Staphylococcaceae OTU; Table 1). Conversely, the common OTU of Corynebacterium was most common in participants who did not use underarm products; they had >109% more Corynebacterium than participants who regularly used deodorant and >335% more Corynebacterium than those who used antiperspirant (Table 1). We examined the evenness of armpit microbes (see Supplemental Information), but there were no significant effects of sampling time, our treatments, or their interaction on this metric of community structure.

Figure 2 Non-metric multidimensional scaling plot of armpit microbes based upon rarefaction using 1,000 sequence reads.

Small symbols represent individuals from each treatment group and large symbols represent group centroids ±1SE.

To further examine patterns in the abundances of Staphylococcaceae and Corynebacterium, mixed model ANOVAs were performed to test for day and gender effects, and interactive effects of these variables with product use on each of these two taxa, both of which were important in our analyses, but are also known to be functionally important armpit taxa. An effect of product was observed (Fig. 3A, F2,11 = 8.36, p = 0.006), such that the abundance of the OTU of Staphylococcaceae was lower in participants who did not use underarm products compared to users of antiperspirant (Tukey HSD, p = 0.005) or deodorant (Tukey HSD, p = 0.007), as expected based on results of the composition analysis (see mean abundance values in Table 1). Conversely, users of no product had significantly higher abundances of Corynebacterium than users of antiperspirant (Tukey HSD, p < 0.001) or deodorant (Tukey HSD, p = 0.006) (Fig. 3B, F(2, 11) = 16.56, p < 0.001). Corynebacterium abundance tends to be positively associated with the strength of body odors (Taylor et al., 2003). This pattern is the opposite of what we would expect if the bacteria in the armpits of product users are different from those of non-product users because product users are often individuals who have more odorous biotas (Harker et al., 2014). No other significant main or interactive effects were observed, including no effect of day on abundances of Staphylococcaceae or Corynebacterium. Whereas Callewaert et al. (2013) found that females tended to be dominated by Staphylococcus spp., and males by Corynebacterium species, we observed no effect of gender on abundances of these two bacterial lineages.

Figure 3 Mean abundances of Staphylococcaceae and Corynebacterium across product use groups.

Mean abundances of (A) Staphylococcaceae and (B) Corynebacterium of participants who regularly used antiperspirant, deodorant or no underarm products based upon sequence data. Underarm product use significantly affected the abundance of both Staphylococcaceae and Corynebacterium (2-way ANOVA: p < 0.0001 for both microbes). However, neither sampling period nor its interaction with product use significantly affected either microbe (2-way ANOVA: p > 0.05).

As a result of the differential associations of these two bacterial taxa to product use, the abundances of Staphylococcaceae and Corynebacterium were strongly negatively correlated to each other across all individuals (Fig. 4; Pearson Rank Correlation: r = − 0.697, p < 0.0001).

Figure 4 Relationship between the abundances of Staphylococceae and Corynebacterium across all individuals.

Spearman rank correlation: r = − 0.697, p < 0.0001.

Discussion

Overall, we found an initial negative effect of antiperspirant, but not deodorant, on bacterial abundance using a traditional culture-based approach. After one day of ceasing product use, antiperspirant users had fewer colonies of culturable bacteria than deodorant users or users of no product. Colony abundance increased, particularly across Days 2–5, with continued product disuse. When all participants began to use antiperspirant, bacterial counts declined. Together these results demonstrate, as expected, that antiperspirants are capable of strongly reducing the biomass of the armpit microbial community, largely independent of the historic product use of individuals. In short, antiperspirant appears to have a clear negative effect on bacterial abundance, one that can be detected in individuals using antiperspirant and that can be produced experimentally. The effect of deodorant on bacterial abundance is more modest, if present at all. These results were not unexpected, in light of two key differences between deodorants and antiperspirants: (1) many deodorants are ethanol-based and likely more water soluble and easier to wash away than antiperspirants; and (2) antiperspirants contain aluminum-based salts that reduce sweat by forming precipitates that physically block sweat glands (Benohanian, 2001) and thus may reduce resources necessary for the growth of microbial communities.

A key question though, given the relative difficulty of culturing many axillary bacteria, particularly those that are slow-growing, is the extent to which changes in the abundance of easily cultured bacteria in response to a short-term experiment match the differences in the entire assemblage, when evaluated using more comprehensive sequencing approaches, particularly differences seen in association with long-term use. Here, several practical challenges exist. Long term experiments on deodorant and antiperspirant, experiments conducted over years, require very committed participants. They are also, however, ethically questionable if prolonged use of deodorants and antiperspirants can have persistent effects on microbial assemblages that in turn, affect health and well-being. In addition, it has proven very challenging to isolate DNA from participants actively using underarm products. Our approach to dealing with this challenge was to use sequence-based approaches to consider the assemblages of microbes in armpits immediately after product use was ceased (as a measure of long-term differences among individuals differing in product use, with only a short period for any shift post disuse). In addition, we considered samples from several days later, once some shift might have been able to occur. These sequence-based data are largely correlational rather than experimental and yet allow strong inference when coupled with experimental data on easily culturable microbes.

Based on sequencing of 16S rDNA, long-term antiperspirant users tended to have more bacterial OTUs in their armpits (multiple days after ceasing product use) than did long-term deodorant users (after disuse of product; Fig. 1). We expected that application of underarm products would negatively affect dominant species, thereby creating more opportunities for rare species to become established. However, we did not observe this effect in deodorant users, who actually had fewer species of bacteria in their armpits compared to armpits of participants who use no product (Fig. 1). Our findings are consistent with those of Callewaert et al. (2014). In line with Callewaert et al.’s (2014) comparison of individuals, we observed a larger change (i.e., increased community richness) in the armpit microbial community when regular product users stopped wearing and then resumed use of antiperspirants, compared to those asked to stop then resume use of deodorants. As such, we expect that our results represent a general effect of antiperspirant use. We can only speculate as to why effects of stopping long-term deodorant and antiperspirant use might have such disparate effects, though note that the particular antiperspirant products (i.e., brands) our participants reported to us all contain aluminum salts. These compounds may alter the underarm habitat (in a manner that deodorants do not), and provide a selective advantage to bacteria not historically common in the human armpit habitat. Based on our study, this underarm habitat alteration lasts multiple days after stopping product use (see Fig. 1, which is based on data from days 2 and 5). The highly abundant microbes identified here compare reasonably well to other studies analyzing moist areas of human skin showing high abundance of Staphylococcaceae and Corynebacteriaceae, among others (see Grice & Segre, 2011 for a review).

The composition of armpit bacterial communities of both antiperspirant and deodorant users was associated with differences in abundances of the two most abundant bacterial taxa, an OTU of Staphylococcaceae and an OTU of Corynebacterium, relative to users of no product. The Staphylococcaceae OTU was the most dominant bacterial group in participants ceasing antiperspirant and deodorant use, followed by the Corynebacterium OTU, whereas this dominance order was reversed among users of no product. In our sequence-based study, we cannot preclude the possibility that individuals who use deodorant and antiperspirant tend to have non-random assemblages of armpit bacteria. But we would expect that, if anything, such individuals would tend to have more odoriferous assemblages of microbes. Armpit odor is largely associated with Corynebacterium, such that we would then expect more Corynebacterium in product users: we found the opposite.

Unlike many taxa on the body, these two taxa have been relatively well characterized with regard to their biology. Species of Corynebacterium are associated with the dominant odors of the armpits and individuals with more Corynebacterium are likely to have stronger body odor (Taylor et al., 2003). Ceasing the use of deodorant and antiperspirant was associated with lower levels of Corynebacterium, in line with expectations, given that companies that sell underarm products aim to reduce body odor through reduction in overall bacterial counts.

We recognize that two additional considerations may have affected our results and those of other studies of armpits. First, because as much as 90% of the bacterial OTUs identified through DNA-sequence based surveys such as ours are bacterial taxa that typically cannot be cultured under standard laboratory conditions, we chose to culture bacteria to inform when to conduct our sequence based analyses (i.e., to determine if residual product impaired colony growth), and to make general comparisons across product use groups. As such, we used standard LB plates grown under aerobic conditions, which like all media, only allow the culturing of a subset of lineages. Although this did not affect the overall conclusions of our sequence-based results, this may have affected the abundances, in that perhaps those bacteria that were abundant in the armpits of non-product users (e.g., Corynebacterium) were not those easily grown on LB agar. However, this does not account for the increased abundance of culturable bacteria in non-product users from days 1 to 4, which is either due to chance and small sample size, or some systemic change that applied to all of our participants.

Second, in comparing the armpit communities of product-users vs. non-product users, we expect that non-product users began our study with more stable armpit communities than those of product-users who recently ceased using product, a standard feature of press experiments, which are common in ecology. Press experiments are designed to understand whether the application of some treatment and then its removal have similar effects (powerful evidence for the influence of the treatment). However, all press experiments provide direct evidence about the experimental effect for the time interval of the study. Our focus was on eight days, sufficient time for many generations of bacteria. It is very possible that were our experiment shorter or longer that our results might have been different. The armpit is a dynamic system and future studies might usefully follow-up with longer term experiments, though only after careful consideration of the ethics of such experiments given that the bacteria whose composition is altered by deodorant and antiperspirant are of direct health consequences (Christensen & Brüggemann, 2014).

A larger sample size would allow us to test hygiene effects of washing frequency and soap type (as these may potentially disturb armpit communities, even of non-product users) as well as additional demographic factors, such as age and gender. This latter point will be informative as gender differences between Staphylococcus and Corynebacterium have been noted in other studies (Callewaert et al., 2013) and will be informative to tease apart gender biases from our product use categories.

Conclusions

Although it has long been recognized that skin bacterial composition varies strongly among individuals, accounting for such variation has been a challenge, one that has led some authors to suggest that the composition of the skin biome might simply be stochastic, a function of chance colonizations and unpredictable dynamics. Here, we find that the composition of the armpit microbiome is highly predictable, being dominated by Staphylococcaceae and Corynebacterium, and strongly influenced by product use. Species of the Staphylococcaceae include beneficial symbionts (Rosenthal et al., 2011; Christensen & Brüggemann, 2014) but also dangerous pathogens (Otto, 2009; Ryu et al., 2014). It is noteworthy in this regard that the armpit is a common site for pathogenic MRSA infections in athletes (Cohen, 2008). However, we cannot discern which of these taxa are being favored with product use based on our data.

The broader health consequences of antiperspirant and deodorant use are not well studied. Although it has been suggested that deodorant and/or antiperspirant use is associated with incidence or age of breast cancer diagnosis (McGrath, 2003), support for this association is equivocal at best (Hardefeldt, Edirimanne & Eslick, 2013). Whether antiperspirant or deodorant tends to favor less beneficial or even pathogenic bacterial species does not seem to have ever been considered. Recent work indicates that the microbial community structure of the skin, including its commensal/symbiotic residents, exerts significant influence on human health and disease, particularly in the emergence of pathogenic strains of Staphylococcus aureus, S. epidermidis, and Propionibacterium acnes (Otto, 2009; Rosenthal et al., 2011; Christensen & Brüggemann, 2014). Rosenthal et al. (2011) hypothesized that the skin microbiome may be “an antibiotic resistance reservoir,” as has been shown to be the case with the human gut microbiome (Sommer, Dantas & Church, 2009). Our work clearly demonstrates that antiperspirant use strikingly alters armpit bacterial communities, making them more species rich. Because antiperspirants only came into use within the last century, we presume that the species of bacteria they favor are not those historically common in the human armpit. Whether these species may interfere with the function of beneficial skin symbionts, contribute antibiotic resistance genes, prove benign, or perhaps even confer beneficial effects to human health remains an intriguing avenue for further study.

Supplemental Information

Table S1 OTU data rarefied to 1,000 reads for each individual

Citizen science identifiers include participant letter code (A–T), Left (L) versus Right (R) armpit, day 2 or day 5 sampling, product category (Anti, antiperspirant; Deod, deodorant; None, no product use), and Female (F) versus Male (M). Cells shaded gray correspond to data from the pit that was not used for downstream sequence analysis. Participants reported how many times they showered during the study; reported per week along the top.

Click here for additional data file.

Table S2 Randomly chosen armpit for sequence analysis

Click here for additional data file.

Table S3 Average abundance of OTUs based on product use group

Note: an asterisk indicates those unique to antiperspirant users.

Click here for additional data file.

Table S4 OTUs differentially abundant between product use groups with a q-value threshold of 0.05

Click here for additional data file.

Supplemental Information 1 Supplemental Material

Evenness information of armpit microbes

Click here for additional data file.

We thank Greg Pahel for assistance with laboratory work. We thank all of the dedicated citizen scientist volunteers who participated in our study. We also thank the reviewers of our manuscript for their time and insight.

Additional Information and Declarations

Competing Interests

Author Contributions

Human Ethics

DNA Deposition

The authors declare there are no competing interests.

Julie Urban performed the experiments, analyzed the data, wrote the paper, prepared figures and/or tables, reviewed drafts of the paper.

Daniel J. Fergus performed the experiments, analyzed the data, prepared figures and/or tables, reviewed drafts of the paper.

Amy M. Savage analyzed the data, contributed reagents/materials/analysis tools, wrote the paper, prepared figures and/or tables, reviewed drafts of the paper.

Megan Ehlers conceived and designed the experiments, performed the experiments, analyzed the data, reviewed drafts of the paper.

Holly L. Menninger conceived and designed the experiments, contributed reagents/materials/analysis tools, reviewed drafts of the paper.

Robert R. Dunn conceived and designed the experiments, contributed reagents/materials/analysis tools, wrote the paper, reviewed drafts of the paper.

Julie E. Horvath conceived and designed the experiments, analyzed the data, contributed reagents/materials/analysis tools, prepared figures and/or tables, reviewed drafts of the paper.

The following information was supplied relating to ethical approvals (i.e., approving body and any reference numbers):

1. North Carolina State University Institutional Review Board for the Use of Human Subjects in Research

2. IRB #1987.

The following information was supplied regarding the deposition of DNA sequences:

Sequence data from both axillae at day 3 and day 6 are deposited in NCBI as part of Bioproject PRJNA281417(http://www.ncbi.nlm.nih.gov/bioproject/).

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
