# Peer review of "The effect of habitual and experimental antiperspirant and deodorant product use on the armpit microbiome"

_PeerJ, doi:10.7717/peerj.1605_

## Round 0.1 · original submission · Major Revisions

Dear Julie Urban, Julie Horvath, and colleagues,

Thank you for submitting your work to PeerJ. The two referees appreciated the work as a substantial addition to the field. However, being very thorough in evaluating several aspects of the work, they had a number of comments concerning some aspects of the experimental design and numerous clarifications of methods, phrasing and constructive advice as to where the precision of descriptions should be improved. In some aspects such as the classification of identified organisms into OTU instead of other taxonomic units the referees concur. One major experimental point that is raised specifically by expert no. 2 is the investigation of the axillary microbiome in the different groups before and later during the antiperspirant use, in particular when the choice of groups is concerned, since the user groups can have different starting microbiota. If some of the requested data or samples would still be available, the authors might consider including these data to improve the significance of their study. One further, less experimental, point, but still important with respect to phrasing, is that it should be made clear that the microbiota of the users were sampled directly after stopping the antiperspirant use and not after a longer intermittent recovery period. So the fastest-growing organisms are supposed to reappear sooner, and it is fair to assume that there is no equilibrium at this early sampling time point yet.

Please thoroughly and carefully address all reviewers’ comments, small and substantial, and provide a point-to-point response letter to these comments before resubmitting your work. Please also provide a track-change version.

On a personal note: the editor, in congruence with expert no. 2, asks the authors to consider altering the title of their work slightly, since “Stinking garden”, however catchy (and title phrases are, to some extent, a matter of personal taste), implies a slightly biased qualification of the subject which is rather not sound in scientific terms. It is quite certain readers will appreciate the work even more with a less catchy and more scientific title - removing the first part of the title altogether and highlighting the novelty point over already published work could improve the significance of the title.

Kind regards,
Christine Josenhans

Reviewer 1 ·

Basic reporting

The manuscript “A Stinking Garden: The effect of habitual antiperspirant and deodorant product use on the armpit microbiome” by Urban et al. is well written and self-contained, but some passages lack precision (see below).

The authors do not state clearly whether ethics approval for this study was obtained, or why formal approval was not necessary. They merely report that the participants were given consent forms, but do not indicate whether consent was actually obtained.

Experimental design

The material and methods section is comprehensive, and overall the methods appear sound. However, the sections on sequence-based sampling and analysis lack some important details:
- The method of OTU clustering and the OTU picking strategy (de novo / open-reference / closed-reference) should be specified.
- Both NMDS and PerMANOVA analyses are usually based on matrices of pairwise sample distances. The method of distance calculation should be specified.
- For some key software tools no versions numbers are provided.

I would also recommend including the template-specific part of the primers used in the methods text.

In the results section for the sequence-based analysis (including figure and table captions), it is in some cases not clear whether the statistical statements are based on OTU data or on the taxonomic classification. It appears as though the terms “taxon”, “phylotype” and “OTU” were used interchangeably. As the level of classification varies from phylum to genus (supplementary table 1), and family- and genus-level data appear to be combined in Fig. 1, 3, 4 and Table 2, taxon-level data would be a very questionable basis for statistical analysis. If all analyses were done on OTU level data, then the corresponding passages need rephrasing. I would also suggest including the OTU identifiers in the figures and the table in addition to the classification. If some analyses really were done on the taxon level then these will need repeating on the OTU level.

Table 1 should also include the data for days 7 and 8.

Validity of the findings

The abundance of culturable bacteria in the no products users is comparatively low at day 1 and increases considerably up to day 4. The authors should discuss why this might be and whether this has any implications for the counts in the other two groups.

Data on the individual OTUs should be included in the supplementary data, maybe in a second table.

The authors mention possible implications of the altered armpit microbiota in antiperspirant users, but they do no discuss how the armpit microbiota in the three groups compares to what is known about human skin microbiota from other body sites. A tool such as Metastats (White et al., 2009) could be used to identify the OTUs significantly more abundant in antiperspirant users, and then these could be discussed in more detail.

Before publication, the 454 data should be submitted to a sequence repository and the accession number(s) or project id(s) should be reported in the manuscript.

--
citation:
(White, J.R., Nagarajan, N., Pop, M., 2009. Statistical methods for detecting differentially abundant features in clinical metagenomic samples. PLoS Comput. Biol. 5, e1000352. doi:10.1371/journal.pcbi.1000352)

Additional comments

I am providing an annotated version of the manuscript in which the passages I refer to are marked.

Annotated reviews are not available for download in order to protect the identity of reviewers who chose to remain anonymous.

Reviewer 2 ·

Basic reporting

Major remarks:
Line 263-266:
“Whereas Callewaert et al. (2013) found that females tended to be dominated by Staphylococcus spp., and females by Corynebacterium species, we observed no effect of gender on abundances of these two bacterial lineages.”
This is material for the discussion. It is females who are dominated by Staph, not the other way around.
Was there an effect of age? Effect of hygiene habits (washing frequence)?


Minor remarks:
Title: “A Stinking Garden” – can refer to odor analysis. The reader might be misinformed by this: odor analyses were not included in the study.

Line 82: Define citizen scientists.

Line 90: typo? drop out --> drop-out

Line 108: Table 1: is it not useful to convert the abundance score into CFU/cm²? This is more meaningful for the reader.

Line 114: Suggest (IBM 2012) --> (IBM Inc., USA)

Line 135: Archaea, with or without capital A?

Line 169: Brackets in brackets are not really preferred.

Line 197-198: Suggest: “no subject”, or “none of the subjects”

Line 199: HSD, please elucidate that this stands for Honestly Significant Difference

Line 199 and onwards: “wearers”, suggest “users”. Wearing refers more to clothes.

Line 204: Table 1:
Did the participants all wash their armpits on Day 0 then?
Suggest conversion to CFU/cm²
The * and a asterics also apply for the three different groups separately, but this is not shown in the Table. Maybe useful to add?
The * and a asterics are also shown on Day 1 and 2. This is maybe confusing. I would suggest to leave the asterics out. Add the information in the Table legend.
Why are the data not shown in the Table for Day 7 and 8? Add them please.

Line 224: suggest reformulation: … “and right armpits for all but 4 samples, as described above),”

Line 228: suggest reformulation: “Samples of bacteria from long-term antiperspirant users x days after stopping the underarm product were more rich …”

Line 256: Figure 3: Unit of Rarified Abundance?

Line 268: negatively correlated to each other? across all individuals

Line 321: typo: Cornyebacterium

Line 344-345: well said!

Experimental design

Major remarks:
Line 86: three groups: Didn’t the authors analyze the participants underarm microbiome on forehand? It would have been practical to divide into three groups based on their dominant axillary bacteria. The selection error could have been excluded by this means.

Line 87-89: Is there information about which deodorant and antiperspirants that were used by the participants? What is the content of these products? Did the participants in one group use one and the same product?

Line 93: Why 8 days? Skin renewal takes about 30 days. The chosen time frame is very short.

Line 94-95: How frequently did the participants apply deodorant or antiperspirant? Once per day? Twice per day? At what time?

Line 95: Only 5 days… What is the rationale for this short time span? How sure are the authors that there is no more underarm product present in the skin or in the sweat glands?

Line 97: The participants resumed with the same product which they normally used?
When did the participants take a bath, shower, or washed their armpits during this period? Or this was not allowed? Was the washing ‘normalized’ according the participants? Which kind of clothes did the participant wore? Every day renewed?

Line 104 - 105: What is the rationale for choosing LB agar plates? Why not blood agar plates (they are better to cultivate axillary bacterial samples)?
Plates were incubated in aerobic conditions? Can the authors add this information? Not that it is needed in the M&M section, but why did the authors choose for aerobic conditions?
Corynebacteria are rather difficult to culture on plate and require facultative or obligate anaerobic environments and specific plates to grow.

Line 123-124: Confusing formulation of “our "early" sequence-based sampling on Day 3 (the second day of continued product disuse) and the "late" sampling on Day 6 (the fifth and final day of continued product disuse)” Suggest reformulation: “we chose to sample on Day 3 (second day of underarm product disuse) and Day 6 (final day of underarm product disuse).”
Why did the authors only sequence on these two moments? Why not choosing for a sample on Day 1 and/or Day 7-8? This would truly reveal the effect of the deodorant and antiperspirant product use. Now the authors can only reveal the effect of the stopping the usage.

Line 159 and onwards: OTU or “phylotype”
An OTU is totally not the same as a phylotype. A phylotype is the sequence identified based on a database (BLAST or RDP or other) and then binned into a group based on the similarity with that database. An OTU is the sequence compared to other sequences in your dataset and binned into groups based on the 97% similarity of other sequences in your dataset. It prevents taking up sequencing errors.
Would recommend to only use the term “OTU”, based on your M&M description.
It is not correct to identify the genus name of an OTU to call it a phylotype. I suggest that the authors took a phylotype for an OTU? This is not correct an can be easily adjusted throughout the manuscript. If the authors would have done another calculation to transform an OTU in a phylotype, then please describe. This would however not be entirely correct. To make use of the term phylotype, the authors need to start from the raw data.


Minor remarks:
Line 82: Is 18 subjects enough?

Line 86: Did the authors obtain an ethical approval as well? (if necessary?)

Line 92: so one participant who used deodorant turned out to use an antiperspirant product?

Line 99: Sampling was done once per day? Every day sampling? Morning, afternoon, evening? Or how many times? It became clear in the results section, but is missing in the M&M.

Line 102: How and how long were the samples stored until further analysis? The swabs were stored dry or in PBS?

Line 102: Add info that samples were taken on every day during the 8 days.

Line 118: Day 1-6: Do the authors mean Day 1 vs Day 2-6?

Line 134: What is the rationale for choosing the V4 region and these primers? Maybe good to mention that you target the V4 region.

Line 163: No evenness analysis? Would be valuable information.

Line 171: Did the authors check the normality of the data?

Validity of the findings

Major remarks:
Abstract: “In addition, individuals who used antiperspirants or deodorants had armpit communities dominated by Staphylococcaceae, whereas those of individuals who habitually used no products were dominated by Corynebacterium. Collectively these results suggest a strong effect of product use on the bacterial composition of armpits. Although deodorants and antiperspirants similarly favor presence of Staphylococcaceae over Corynebacterium, their …”
Be careful with this statement. From these results this can be easily suggested. However, there is a big impact of participant selection in the no underarm product group. If the no underarm product group would have dominances of Staph, the conclusion would be totally different. Many people who don’t use underarm products are dominated with Staph. This statement also implies that people should start using underarm cosmetics, while for many people this is not necessary. Also, the authors did not analyze the armpit microbiome when underarm products were effectively used. So the statement cannot be made and should be reformulated.
In the formulation, it should be mentioned somewhere that the underarm product use was stopped for x days on the moment of analysis.
The abstract should maybe say how long the underarm cosmetic was used or not used.
Also please adjust this information in the (end of the) Discussion.

Line 231: Figure 1:
Data from Day 3 or Day 6?
Would have been very interesting to see the results from Day 1 for the three different groups.
The Figure can be easily misinterpreted: in fact, all results reflect data when no underarm cosmetics were used. Suggest: instead of “Antiperspirant”, use “ceasing long-term antiperspirant use” and instead of “Deodorant”, use “ceasing long-term deodorant use”. Also suggest to change Figure title: “Mean composition and richness of bacterial taxa across product use groups x days after stopping the underarm product”
The molecular data do not reflect the underarm microbiome when underarm cosmetics are used, but when underarm cosmetics were stopped. This should be made clear in the abstract and throughout the manuscript. The data of Callewaert et al. (2014) show that the staphylococci were the first bacteria to regrow when the use of antiperspirants was stopped. Consequently, one cannot say that the use of AP leads to a dominance of Staph. This should be: the stopping of a long-term use of AP leads to a dominance of Staph.
When stopping the underarm product, you open up a complete niche for bacteria. The fastest growers will be abundant first. Staphylococci are known to grow fast. Corynebacteria are fastidious bacteria and need a lot of time to grow.

Line 242: Table 2:
Unit of average abundance?
Day 3 or Day 6?
On Day 3, did the participants wash their axillae in between Day 1 and 3? (so, are the AP ingredients washed away or not?)
How were the axillae washed? Soap, water, shower gel, other? The same for all participants?

Line 250: “>260% more Corynebacterium”
These data suggest that this is caused by the antiperspirant usage... This is actually due to the participant selection in the study design. One can also select staph-dominant participants who do not use underarm cosmetics. The results would have been totally different. Would the authors then say that the usage of deodorants and antiperspirants leads towards less staphylococci?

Discussion: Why did antiperspirant users have lower abundances of bacterial colonies on plate than deodorant users? Can the authors compare this with the ingredient list?
There is another important reason missing in the discussion: the fact that the aluminum salts are toxic to bacteria. They are added in high quantities in antiperspirants. These, and other ingredients, prevent the growth of bacteria in the axillary vault.

Additional comments

Very interesting study. Well written, a bit short discussion.
I am missing valuable information about the washing of the axillae. I have no idea when the axillae were washed, or not washed at all, during the 8 days.
It is a pity that the 454 sequencing was not done on a moment when deodorants or antiperspirants were effectively used in the axillae. This study provides information about the recolonization after a long-term use of underarm cosmetics and cannot judge if the usage of the underarm products favor certain bacterial types. Please revise this carefully.
The selection of the non-deodorant/antiperspirant group was also badly chosen, to my opinion…

---

## Round 0.2 · Major Revisions

Dear authors,

we have received two well-funded and detailed comments regarding some points in the revised version of your manuscript. The authors agree that the revised version has been much improved. They both still find different shortcomings regarding statistics, experiments' and analyses' descriptions/reproducibility/study design which are very important for making the article fully accessible to readers.

All these reviewers' comments are pertinent and it will be essential to address all of them fully, first of all by accurately describing what was done.

Since the reviewers' remaining comments are so important to improve the manuscript, although it'll be probably quite straightforward and fast to address all of these, the decision is earmarked as a major revision.

We are grateful for the thorough revision you have already performed and for the improvements yet to be made. .

Best wishes,
Christine Josenhans

Reviewer 1 ·

Basic reporting

The manuscript has been considerably improved.

Experimental design

While the design and analysis methods are described much better now, I noticed a potentially major point of concern: From the Materials and Methods part, it appears as if the individual samples were subsampled to 1000 sequences, but then pooled so that most sequence sets contained 2000 sequences while others contained only half as many. For most reported analyses it is unclear whether they were performed on a per-sample-basis or on the basis of these unequal sequence sets. Some analyses are apparently based on pooled data even though they are highly dependent on total sequence numbers, such as the analyses of abundances in Fig. 3 and Table 1, or the richness analysis.
The authors should make sure to base all sample size sensitive analyses on equal sequence numbers, or to modify their analyses to accommodate unequal sample sizes. Depending on the analysis, this could be done by averaging across both armpits, performing analyses on individual samples instead of pools, or working on percentages instead of count data. Another option would be to pool the sequences from both armpits before discarding the low-yield samples and then to subsample on the level of study participant. For all analyses, the underlying dataset should be clearly stated.

Validity of the findings

No comments

Additional comments

The manuscript has been considerably improved. However, the concern mentioned above is important enough that I would like to see the manuscript again after revision.

Reviewer 2 ·

Basic reporting

Thank you for this great improvement. But, I do have some additional comments…

Line 80, Line 277-279, Line 363-364: “we compare the abundance of two focal taxa, an OTU categorized as Staphylococcaceae and an OTU of Corynebacterium”
Did the authors look at one single OTU of Staphylococcaceae or also the OTU of Staphylococcus?

Line 92-95 & Suppl Table 2:
Authors mention equal groups of 6 subjects of no product, antiperspirant and deodorant use. In reality there were 5 persons in the deodorant group and 7 in the antiperspirant group. Can this information be added here? Can the authors add the gender of the participants in each group?
Could the fact that the antiperspirant group contained 4 females have affected the final conclusions?
In Supplemental Table 2, is there one person missing of the no product use group? I count 5 subjects of no product use (2 female, 3 male).

Discussion & Suppl Table 2:
With the data in supplemental table 2 the authors could say something about the inter-individual differences (as discussed in Introduction), the differences between left and right armpit and the differences over time. This is optional of course, but would be informative.

Line 105: All participants indicated showering/bathing 3-9 times per week, and they were asked to continue their normal showering/bathing routine for this experimental week
Is it possible to show in supplemental Table 2 some metadata of the participants, like showering frequency and/or antiperspirant/deodorant usage frequency (if available), so the reader can compare this with the results?
Maybe also include the richness and evenness of the results in this table?

Table 2: data are now expressed as CFU/cm².
Are the authors sure that only 1-10^1 CFU/cm² are found in the axillary region? Or do you have to add 10^x?

Table 2:
Results are very low for deodorant user group and no product user group. Can you elaborate on this?
The no product user group shows an increase in bacterial presence from Day 1 to 4. Can this be related to the washing habits? Can you look into the individual data for this? It would be handy to see the individual data in supplementary information.

Line 304: “we observed no effect of gender on abundances of these two bacterial lineages”
When looking into the data in Suppl Table 2, it can be counted that females overall have 53% staphylococci and 23% corynebacteria. Males have 44% staphylococci and 40% corynebacteria. Are the authors sure that there is no significant difference between the male and female group?

Line 419: When Corynebacterium become more rare, as we show occurs via habitual use of underarm products…
I’m sorry, but I cannot agree with this sentence. How did the authors prove that corynebacteria become more rare by using more underarm products?

Figure 4: please add the units to the axis.

Line 174: de novo --> italic?
Line 80: typo: Staphycloccaceae --> Staphylococcaceae
Line 97 & 110: typo: Aluminum zirconium tricholohyrex Gly --> aluminum zirconium trichlorohydrex Gly
Line 251: Corynebacteria. --> without capital C, not italic (it is a trivial name)
Line 304, 408: typo: Staphylococcus --> italic
Line 371: typo: Corynebaterium, --> Corynebacterium,
Line 326-330: rather long sentence – difficult to read/comprehend.
Line 432: typo: Staphalococcus aureus --> Staphylococcus
Legend Figure 4: typo: Staphylococceae --> Staphylococcaceae
Please carefully review for other typo’s.

Experimental design

see above

Validity of the findings

see above

---

## Round 0.3 · accepted · Accept

Dear Dr. Urban,

thanks for revising your manuscript carefully. For Table 2, for reasons of mathematical accuracy, it is more accurate to really multiply all the counted bacterial numbers by 25, as you also suggested in your response to reviewer. I would strongly recommend to do this throughout the table to avoid misunderstandings, since the data is given by cfus (numbers) per cm2 and not per 20 nl plated material.

It did also not become entirely clear at all places in the text, whether the bioinformatics for subsampling of microbiota and subsequent analysis was always to the same numbers of sequence reads for all samples (although it becomes clear now that the same type of non-pooled samples was used for the experimental samples). Please make sure to have this clarified as well in the methods and throughout he text, since it affects the interpretation of the analyses.

Otherwise, all open questions have been fully answered now. Thank you for submitting such a fine work to PeerJ.